# High Expression of IRS-1, RUNX3 and SMAD4 Are Positive Prognostic Factors in Stage I–III Colon Cancer

**DOI:** 10.3390/cancers15051448

**Published:** 2023-02-24

**Authors:** Hallgeir Selven, Lill-Tove Rasmussen Busund, Sigve Andersen, Mona Irene Pedersen, Ana Paola Giometti Lombardi, Thomas Karsten Kilvaer

**Affiliations:** 1Department of Oncology, University Hospital of North Norway, 9038 Tromsø, Norway; 2Department of Clinical Medicine, UiT The Arctic University of Norway, 9038 Tromsø, Norway; 3Department of Pathology, University Hospital of North Norway, 9038 Tromsø, Norway; 4Department of Medical Biology, UiT The Arctic University of Norway, 9038 Tromsø, Norway

**Keywords:** colon cancer, RUNX3, SMAD4, biomarker, prognosis

## Abstract

**Simple Summary:**

We studied the expression of several protein biomarkers in both stromal and tumor tissue from colon cancer patients. High expression of IRS1 in stromal tissue and RUNX3 and SMAD4 in both stromal and tumor tissue were positive prognostic factors. Of particular interest, RUNX3 expression in stromal tissue was associated with the density of tumor-infiltrating lymphocytes. This finding may be important for understanding the prognostic impact of lymphocytes and predicting and increasing the efficacy of immunotherapy in colon cancer.

**Abstract:**

Colon cancer is a common malignancy and a major contributor to human morbidity and mortality. In this study, we explore the expression and prognostic impact of IRS-1, IRS-2, RUNx3, and SMAD4 in colon cancer. Furthermore, we elucidate their correlations with miRs 126, 17-5p, and 20a-5p, which are identified as potential regulators of these proteins. Tumor tissue from 452 patients operated for stage I–III colon cancer was retrospectively collected and assembled into tissue microarrays. Biomarkers’ expressions were examined by immunohistochemistry and analyzed using digital pathology. In univariate analyses, high expression levels of IRS1 in stromal cytoplasm, RUNX3 in tumor (nucleus and cytoplasm) and stroma (nucleus and cytoplasm), and SMAD4 in tumor (nucleus and cytoplasm) and stromal cytoplasm were related to increased disease-specific survival (DSS). In multivariate analyses, high expression of IRS1 in stromal cytoplasm, RUNX3 in tumor nucleus and stromal cytoplasm, and high expression of SMAD4 in tumor and stromal cytoplasm remained independent predictors of improved DSS. Surprisingly, with the exception of weak correlations (0.2 < r < 0.25) between miR-126 and SMAD4, the investigated markers were mostly uncorrelated with the miRs. However, weak to moderate/strong correlations (0.3 < r < 0.6) were observed between CD3 and CD8 positive lymphocyte density and stromal RUNX3 expression. High expression levels of IRS1, RUNX3, and SMAD4 are positive prognostic factors in stage I–III colon cancer. Furthermore, stromal expression of RUNX3 is associated with increased lymphocyte density, suggesting that RUNX3 is an important mediator during recruitment and activation of immune cells in colon cancer.

## 1. Introduction

The colon comprises one fifth of the digestive tract’s length. However, despite its small lengthwise contribution, the colon harbors more than 40% of digestive tract cancers. With an estimated incidence of 106,000, colon cancer will be the fifth most common malignancy in the USA in 2022. Moreover, it will be the second most common cause of cancer-related deaths [1]. Despite the increasing knowledge of colon cancer etiology, patients’ prognoses have not improved significantly over the last decade. Consequently, novel prognostic and predictive biomarkers are needed to (1) identify patients at high risk of colon cancer death and (2) to select the right candidates for adjuvant and novel treatments.

In our previous works, we found that high expression levels of miRs 126, 17-5p, and 20a-5p were positive prognosticators in early stage colon cancer patients [2,3]. In an attempt to further elucidate the role of miRs 126, 17-5p, and 20a-5p, we chose to focus on IRS-1 and 2, RUNX3, and SMAD4. These proteins are four members of three distinct protein families thought to be regulated by these miRs [4,5,6]. They are directly involved in important signaling pathways including the ERK, PI3K/AKT, TGF-β pathways, among others [7,8,9,10]. Moreover, they are already implicated in colon cancer development through cell line experiments and a few prognostic studies [11,12,13]. IRSs are important in insulin signaling and maintain cellular functions such as growth, survival, and metabolism [14]. There are six known IRS-substrates, IRS-1 to IRS-6, of which IRS-1 and IRS-2 are widely expressed in humans [15]. IRSs can be oncogenic and induce malignant transformation. In addition, IRSs are required to facilitate the transforming ability of other oncogenes, depending on IRS tyrosine phosphorylation. IRS-1 overexpression with subsequent ERK 1/2 pathway activation in fibroblasts was shown to promote cellular proliferation. Previous studies have shown that IRS-1 and 2 are overexpressed in hepatocellular carcinoma (HCC) and pancreatic cancer [16,17,18,19]. However, IRS-1 expression was decreased in squamous cell lung cancer. Moreover, IRS-1 is constitutively activated in several sarcomas and in breast cancer [20], whereas IRS-2 is constitutively activated in patients with the hereditary condition multiple endocrine neoplasia type 2 (MEN2) [10].

RUNX3 is a protein belonging to the runt domain family of transcription factors involved in mammalian developmental pathways [21]. The RUNX3 gene is localized at the 1p36 locus, a region believed to harbor one or more tumor suppressor genes, as loss of heterozygosity in this region is observed in gastric, breast, ovarian, and colon cancers [22]. Activation of TGF-β phosphorylates SMAD3, which associates with SMAD4 and enters the nucleus. In the nucleus, SMAD3/SMAD4 forms complexes with RUNX3, thereby mediating the suppressive effects of TGF-β [8]. The tumor suppressive effect of RUNX3 can be inactivated in several ways (mutations, methylation-related transcriptional silencing, and mis-localization to the cytoplasm) [23]. Furthermore, RUNX3 is an important mediator during the development of both CD4+ and CD8+ cytotoxic T-lymphocytes (CTL), and is thus likely pivotal in the development of anti-tumor immunity [24]. The polarization of CD4+ towards CTLs may be especially important in the gut epithelium [25].

SMAD is a family of transcription factors, acting as mediators of the TGF-β signaling cascade. There are three functional classes of SMAD proteins in mammals: the receptor-regulated SMADs (SMAD1, 2, 3, 5, and 8); the co-mediator SMAD (SMAD4); and the inhibitory SMADs (SMAD6 and 7). The TGF-β/SMAD4 signaling pathway controls a wide range of cellular processes including proliferation, differentiation, apoptosis, migration, and cancer initiation and progression [26]. In early tumorigenesis, the TGF-β/SMAD4 signaling pathway acts as a tumor suppressor, inducing cell cycle arrest and apoptosis. However, as cells progress through tumorigenesis, they become refractory to TGF-β-mediated growth inhibition and respond by stimulating pathways resulting in TGF-β-mediated tumor progression. Numerous other pathways interact with the TGF-β/SMAD4 pathway, including MAPK, ERK, and PI3K/AKT [27,28,29]. SMAD4’s role in cancer was first discovered in 1996, where SMAD4 gene alterations were shown to be closely related to pancreatic cancer [30]. Loss of SMAD4 has also been reported in cholangiocarcinomas and colorectal cancer, among others [31,32].

In this study, we apply deep learning to digitized pathology images to explore compartment level expression and prognostic impact of IRS-1, IRS-2, RUNX3, and SMAD4 in resected tumors from 452 colon cancer patients. Furthermore, we elucidate their correlations with the expression of miRs 126, 17-5p, and 20a-5p as well as the density of CD3+ and CD8+ tumor-infiltrating lymphocytes.

## 2. Materials and Methods

### 2.1. Study Population

Patients undergoing radical surgery for colon cancer in various hospitals in Northern Norway in the time period of 1998–2007 were eligible for inclusion. From an initial 861 identified patients, 452 patients were finally included in the study. The main exclusion criteria were metastatic disease/prior malignancy within the last 5 years before diagnosis, missing tissue blocks/inadequate tissue for TMA construction, and faulty coding (rectal cancer, mainly). Follow-up was completed on 1 December 2017. The study population was previously published in detail [2].

### 2.2. Tissue Microarray Construction

All colon cancer cases were reviewed by two pathologists. The most representative areas of tumor without necrosis were selected. Using a 0.6 mm diameter stylet, a total of 4 cores were sampled for each included patient, securing both tumor and stromal tissue. The TMAs were assembled using a tissue-arraying instrument (Beecher Instruments, Silver Springs, MD, USA). The detailed methodology has been previously reported [33]. Sections were cut on a MICROM HM 335 S microtome, transferred to Super Frost Plus slides, and dried at room temperature before staining.

### 2.3. Immunohistochemistry and In Situ Hybridization

The Discovery Ultra Research instrument Roche 05987750001 was used to examine the protein expression of the six biomarkers in colon cancer TMAs. The antibodies used for this study were IRS1 ab40077, IRS2 ab134101, SMAD4 ab40759, and RUNX3 ab135248, sourced from Abcam. In addition, we used CD3 (2GV6) Roche 05278422001 and CD8 (SP57) Roche 05937248001 for in vitro diagnostic (IVD) use. All antibodies were validated for IHC-P (formalin fixed and paraffin embedded tissue) by the supplier. Optimization of dilutions, incubation times and temperatures were done in-house. Advised positive tissue controls from supplier were tested for each antibody. Staining and antibody specificity was verified by an internal tissue control (TMA multi control) containing several normal and cancer tissues. Negative controls were conducted by omitting the primary antibody, The negative controls were mainly clean, but weak brown non-specific staining in RUNX3 DAB stain was observed. Details of the optimized IHC protocols are given in Appendix A. Appendix A shows product information of antibodies and reagents. The methodology for in situ hybridization of miRs 126, 17-5p, and 20a-5p was previously described [2,3].

### 2.4. Digitization/Immunohistochemistry Scoring

TMA slides were digitized using a Pannoarmic Flash III digital slide scanner (3DHistech, Budapest, Hungary) and processed in QuPath vs. 0.3.2 according to Bankhead et al. [34]. Only cores containing tumor tissue were used for analyses. Cells were identified and classified using a StarDist deep learning (DL) model trained on the hematoxylin channel of the Lizard dataset [35,36,37]. Briefly, we extracted the hematoxylin optical density image channel, normalized it, and applied the DL model. As an output, we obtained segmented nuclei classified into six classes (neutrophil, epithelial, lymphocyte, plasma, eosinophil, and connective). Due to limitations with the DL method, lymphocytes, plasma cells, eosinophils and connective tissue cells were combined into a single stromal class. The final analyses were conducted on the tumor epithelial cells and the combined stromal class cells. In addition, minor filtering based on size and circularity was applied. The mean marker intensity and the estimated cytoplasm (an arbitrary expansion from the nucleus) were then calculated separately for each nucleus. The final score for each compartment is the median score of all its nuclei. QuPath scripts to run and generate the final scores for each marker are available upon request to the corresponding author. An optimal cutoff strategy was applied to dichotomize markers for survival analyses. For non-significant markers, the median cut-off was chosen.

### 2.5. Statistics

Statistical tests were performed in Rstudio 2021.09.0 build 351 (RStudio PBC) using R version 4.0.4. Disease-specific survival (DSS) was defined as the interval from surgery to the time of colon cancer death. Before analyses, expressions of the investigated markers were rescaled to a range between 0 and 1 using max–min scaling. Kaplan–Meier plots visualized the dichotomized molecular marker’s impact on patient survival. The differences between the survival curves were tested using the log-rank test. Multivariable analyses were conducted using Cox regression. All significant variables from the univariate analyses available to a majority of the patients were entered into the initial models. A sequential backward conditional approach was adapted, where variables with p-values above 0.1 were dropped at each step. Chi-squared and Fischer’s exact tests were used to examine the association between molecular marker expression and clinicopathological variables. Pearson correlation was used to examine the associations between marker expressions. *r* values of ±0, 0–0.2, 0.2–0.3, 0.3–0.5, 0.5–0.7 and >0.7 were considered negative, weak, weak/moderate, moderate, moderate/strong, and strong, respectively. Hierarchical clustering with distance calculated based on the *r* values of their correlations was applied to visualize patterns in the correlation data.

## 3. Results

### 3.1. Patient Characteristics

The patients characteristics were previously reported [2]. A total of 452 patients were included in the study. All patients were treated with curative intent, and most patients were diagnosed with pTNM stage II (48.5%) and III (35.6%). Survivors were followed up on for a median of 173 months. At the end of follow-up period, 119 patients experienced a recurrence and 108 had succumbed to their disease.

### 3.2. Expression of SMAD4, RUNX3, IRS-1, and IRS-2 and Their Correlations with Clinicopathological Variables

Expression of the investigated markers are illustrated in Figure 1. As can be seen, SMAD4 and RUNX3 were expressed both in nucleus and cytoplasm whereas IRS-1 and IRS-2 expression was restricted to the cytoplasm. IRS-1 and 2 and SMAD4 expression was evenly distributed in tumor epithelial cells, spindle shaped cells/stromal cells, and immune cells. RUNX3 was predominantly expressed in tumor infiltrating lymphocytes (TILs) and to a lesser extent in other cells. IRS-1 and 2 were in some cases highly expressed in collagen-like sheets.

Correlations between investigated biomarkers and clinicopathological variables are presented in Appendix A. Expression of SMAD4 in tumor cytoplasm was correlated with weight loss and pathological stage, whereas SMAD4 in stromal cytoplasm was correlated with pathological stage only. SMAD4 in stromal nucleus was correlated with site and histological grade. Expression of RUNX3 in tumor nucleus and cytoplasm were both correlated with site. Expression of RUNX3 in stromal nucleus and cytoplasm were both correlated with site, and RUNX3 in stromal nucleus was correlated with histological grade, whereas RUNX3 in stromal cytoplasm was correlated with pathological stage. Expression of IRS-1 in stromal cytoplasm was correlated with ECOG status, and IRS-2 in stromal cytoplasm was correlated with weight loss.

### 3.3. Correlations between Investigated Biomarkers and CD3, CD8, miR-17-5p, miR-20a-5p and miR-126

Correlations between investigated markers, miRs, and lymphocyte markers are presented in Figure 2. As expected, moderate to strong correlations were observed between epithelial and stromal expression for each marker. Furthermore, extensive correlations between IRS-1 and IRS-2, RUNX3 and SMAD4 in tumor and stroma, were observed (0.15 < *r* < 0.60). S-miR-126 showed a weak/moderate correlation with SMAD4 (0.20 < *r* < 0.30) and S-miR-126 and S-miR-17-5p showed weak/moderate correlations with CD3+ and CD8+ TIL density (0.15 < *r* < 0.30). No other relevant correlations between miRs and other investigated markers were observed. Of note, CD3+ and CD8+ TILs density showed moderate/strong correlations with S-RUNX3 (0.35 < *r* < 0.60) and weak/moderate correlations with S- and T-SMAD4 (0.15 < *r* < 0.30).

### 3.4. Univariate Analyses

Univariate survival analyses of clinicopathological variables in this cohort have been reported earlier, showing that age, weight loss, pathological stage, histological grade, vascular infiltration, and resection margins were significant indicators of DSS [2]. Univariate analyses of investigated biomarkers are presented in Table 1 and Figure 3. Increased expression of SMAD4 in nucleus and cytoplasm, the tumor epithelial compartment, and cytoplasm in the stromal compartment; RUNX3 in the nucleus or cytoplasm in both the tumor epithelial and stromal compartments; and IRS-1 in the stromal cytoplasm were all significant predictors of a favorable DSS.

### 3.5. Multivariate Analyses

Multivariable analyses are summarized in Table 2. Increasing age at diagnosis, pathological stage III, and a resection margin of <1 mm were unfavorable predictors of DSS. Furthermore, tumor epithelial expression of SMAD4 in cytoplasm (HR 0.58, 95% CI 0.43–0.80, *p* < 0.001) and RUNX3 in nucleus (HR 0.62, 95% CI 0.45–0.84, *p* = 0.002) were independent positive predictors of DSS. In the stromal compartment, IRS-1 in cytoplasm (HR 0.64, 95% CI 0.47–0.87, *p* = 0.005), SMAD4 in cytoplasm (HR 0.67, 95% CI 0.5–0.91, *p* = 0.009), and RUNX3 in cytoplasm (HR = 0.62, 95% CI 0.44–0.87, *p* = 0.006) were independent predictors of a favorable DSS.

### 3.6. Co-Expressions

Co-expression analyses between SMAD4 in tumor epithelial and stromal cytoplasm and RUNX3 in tumor epithelial nuclei and stromal cytoplasm are presented in Appendix A. Patients with increased/preserved SMAD4 and RUNX3 expression in either the tumor epithelium (HR = 0.34, 95% CI 0.18–0.66, *p* = 0.001) or the stromal compartment (HR = 0.34, 95% CI 0.2–0.57, *p* < 0.001) had significantly better prognoses compared to those with decreased/lost expression.

## 4. Discussion

To our knowledge, this is the first study to separately investigate tumor epithelial and stromal cell expression of IRS-1 and 2, SMAD4, and RUNX3 in colon cancer patients. We demonstrate that increased expression of RUNX3 and SMAD4 in tumor epithelial and stromal compartments and IRS-1 in stroma are independent predictors of a favorable prognosis in this patient group (Table 1 and Table 2 and Figure 3). The largest difference in five-year DSS was observed for high vs. low RUNX3 expression in stromal cell cytoplasm (84% vs. 63%). Moreover, we have previously assessed the prognostic impact of miRs 126, 17-5p, and 20a-5p expression in tumor epithelial and stromal tissue from early stage colon cancer patients and observed that the investigated miRs were positive prognostic factors of DSS [2,3]. According to several cell line studies, the investigated markers are key players in pathways regulated by these miRs [7,38,39]. Surprisingly, with the exception of weak correlations (0.2 < *r* < 0.25) between miR-126 and SMAD4, no relevant correlations between the miRs and the investigated protein markers were observed. However, moderate to strong correlations were observed between the CD3+ and CD8+ TIL density and stromal cell RUNX3 expression.

Our results corroborate previous observations that RUNX3 is an important tumor repressor in colorectal cancer. In a large trial comprising 849 stage I–IV colorectal cancer patients, Soong et al. demonstrated that RUNX3 expression in the nucleus and not in the cytoplasm was a positive prognostic factor [39]. Shin et al. demonstrated that hypermethylation of RUNX3 was associated with an unfavorable prognosis in a small CRC cohort comprising 62 patients, indicating an inactivated form because of transcriptional silencing [40]. Ogino et al. found that patients with metastatic, microsatellite stable colorectal cancer with hypermethylated RUNX3 had an impaired prognosis when receiving combination chemotherapy compared to patients with unmethylated RUNX3 [41]. Furthermore, hypermethylation of RUNX3 is part of a panel of markers defining the CpG island methylator phenotype (CIMP) of colon cancer [42]. Berg et al. documented an OR of 3.4 for local recurrence for patients with loss of RUNX3 expression in a small series comprising 64 stage II–III colon cancer patients evaluated for MSI, CIMP, and copy number variation [43]. Soong et al. was the only other study showing differentiated expression of RUNX3 between tumor cell nuclei and cytoplasm [39]. In addition to prognostic significance, they demonstrated a reciprocal relationship between nuclear and cytoplasmic expression of RUNX3 and stage. Intriguingly, patients with neither nuclear nor cytoplasmic expression showed similar survival to patients with only nuclear expression. In our material we observed a trend towards impaired prognosis for patients with low nuclear and high cytoplasmic expression of RUNX3 in the tumor epithelial compartment. However, using our method and cut-offs, it is not possible to conclude that this staining represents exclusive cytoplasmic expression in tumor epithelial cells and not confounding immune cell expression. Kim et al. used a combination of preclinical models and retrospective data to investigate the connection between RUNX3 and headgehog (Hh) signaling in CRC [44]. Similar to us and others, they found that loss of RUNX3 indicated a poor prognosis. Furthermore, they observed that RUNX3 was a negative regulator of GLI1, the main activator of genes regulated by the Hh pathway. These results suggest that RUNX3 is strongly involved in regulating the Hh pathway in CRC [44]. Moreover, and of particular interest, we found that high stromal cell expression of RUNX3 was associated with favorable survival and was moderately to strongly correlated with CD3+ and CD8+ TILs density. We were not able to identify previous studies investigating compartment specific expression of RUNX3 in CRC. RUNX3 is an important mediator during lymphocyte differentiation into CD4+ and CD8+ cytotoxic T -lymphocytes (CTL) and natural killer cell progenitors into natural killer (NK) cells, and is thus likely pivotal in the development of anti-tumor immunity [24]. Furthermore, data from inflammatory bowel disease and preclinical models suggest that RUNX3 induced polarization of CD4+ cells towards a CD4+ CTL phenotype may be especially important in the gut epithelium [25,45,46]. Although our experimental data suggest a link between lymphocyte expression of RUNX3 and prognosis, we were not able to assert the stromal RUNX3 signal to specific cell types due to limitations with our method. Nevertheless, our results strongly suggest RUNX3 in conjunction with TILs and/or other immune cells as drivers of the favorable survival we observe in the stromal RUNX3 high group.

SMAD4 is a known positive prognostic biomarker in CRC [11]. In our univariate analyses, patients with high expression of SMAD4 in either the tumor epithelial (HR 0.5 for both nucleus and cytoplasm in univariate analyses, and 0.62 for cytoplasm in multivariate analyses) or the stromal compartment (HR 0.5 and 0.67 for cytoplasm in univariate and multivariable analyses, respectively) exhibited beneficial prognosis vs. patients with low expression. These results are in line with those presented by Voorneveld et al. in a large meta-analysis of SMAD4 in CRC in 2015 [47]. Furthermore, supporting the robustness of our findings, they reported the percentage of preserved SMAD4 across the studies to be 50–90%, which corresponds favorably to our chosen cutoff, where approximately 75% of patients are in the group with high SMAD4 expression. Interestingly, in a recent study on patients resected for liver metastases, Kawaguchi et al. reported that the detrimental effect of RAS alterations was abrogated if both TP53 and SMAD4 was preserved [48]. Furthermore, the small group with concurrent RAS, TP53, and SMAD4 alterations exhibited the worst prognosis in both resectable and unresectable patients. These findings suggest that SMAD4 expression may be used to determine whether patients with limited liver metastases are likely to benefit from metastasectomy. We found SMAD4 expression to be a positive prognostic factor in the stromal compartment. Surprisingly, we were only able to identify one study that investigated the expression of SMAD4 in stroma. Contrary to our findings, Bacman et al. did not find any correlations between loss of SMAD4 in neither the tumor, nor the stromal compartment, in a study of 310 stage II–III CRC patients [49]. However, similar to our results, loss of SMAD4 in tumor and stroma was correlated with high grade tumors. Interestingly, Mesker et al. found a positive correlation between preserved SMAD4 and the total amount of tumor associated stroma [50]. Their results indicate that tumor SMAD4 signaling interacts with the tumor micro-environment. Furthermore, at least one preclinical study suggests that preserved SMAD4 signaling is necessary for T-cell suppression of CRC development [51]. Based on weak to moderate correlations with tumor epithelial and stromal RUNX3, IRS-1, CD8+ TIL density, and stromal miR126 (Figure 2), stromal SMAD4 expression in colon cancer is likely involved in complex interactions between tumor epithelial cells and different types of stromal cells. However, further studies are needed to draw conclusions on the role of stromal SMAD4 expression in colon cancer. SMAD4 and RUNX3 are both integral players in the TGF-β pathway [7,8]. In our material, SMAD4 and RUNX3 were weakly to moderately correlated both between and within the tumor epithelial and stromal compartments (0.3 < *r* < 0.6, Figure 2). Hence, it was pertinent to test their compartment specific co-expressions (Appendix A). Not surprisingly, patients with a SMAD4-/RUNX3- pattern exhibited the worst prognosis in both compartments (5-year DSS 60%). Interestingly, preserved SMAD4 or RUNX3 resulted in similar 5-year DSSs in the tumor epithelial compartment ( 70%), but not in the stromal compartment (SMAD4-/RUNX3+ 80%, SMAD4+/RUNX3- 60%). The latter indicates a more important role for RUNX3 in the stromal compartment. These co-expression analyses, combined with our observation of RUNX3 being predominantly expressed in TILs, indicate that RUNX3 plays an important role in TIL-mediated colon cancer suppression.

Numerous studies in other cancer types have shown the oncogenic capacity of IRS-1 and 2, but little is known of their prognostic impact in colorectal cancer [16,17,18,19]. Interestingly, data from epidemiological studies suggest that specific genetic polymorphisms in the IRS-1 and 2 genes are associated with decreased or increased risk of developing colorectal cancer [52,53]. However, we did not observe any association between tumor expression of IRS-1 and 2 and survival in our cohort. In the stromal compartment, IRS-1 expression was an independent positive prognosticator. Stromal IRS-1 was moderately (0.4 < *r* < 0.6) correlated with tumor epithelial and stromal SMAD4, and tumor epithelial and stromal RUNX3, indicating a possible link between stromal IRS-1 and pathways involving activated SMAD4 and RUNX3.

## 5. Future Works

As a notion for future works, we would like to outline the following. (1): S-RUNX3 was predominantly expressed in lymphocyte-like cells. However, morphological classification is prone to error, especially when it is largely based on the hematoxylin staining alone. Further studies should explore S-RUNX3 expression using multiplex immunohistochemistry or fluorescent labeling differentiating specific cell types. We would start with the pan T-lymphocyte marker CD3 and expand the panel of markers if this initial co-expression does not fully explain our current findings. Furthermore, as different types of immune cell activation must be considered as a pan-cancer event, future studies may also include other cancer types. (2): The interactions between T-SMAD4/T-RUNX3 and S-SMAD4/S-RUNX3 warrants further scrutiny in a clinical setting. Even though pre-clinical studies suggest that their silencing results in a deregulation of the TGF-β pathway, clinical confirmation is needed, as other pathways may be involved in this complex interplay.

## 6. Conclusions

We present the first study that differentiates between expression of IRS-1 and 2, RUNX3 and SMAD4 in the tumor epithelial and stromal compartments of colon cancer patients. We confirm previous studies reporting preserved RUNX3 and SMAD4 in the tumor epithelial compartment as positive prognosticators in colon cancer. Furthermore, we present novel data on the positive prognostic value of stromal RUNX3, SMAD4, and IRS-1 in colon cancer. In addition, we show that stromal RUNX3 is correlated with TIL density in colon cancer. Not surprisingly, co-expression analyses indicate that RUNX3 plays an important role in TIL-mediated colon cancer suppression. Clinical implications of loss of RUNX3 and/or SMAD4 expression in the tumor epithelial compartment may be used to identify stage II patients in need of adjuvant chemotherapy, or patients with T1-3N1 tumors eligible for extended chemotherapy compared to the three month standard for this risk group. However, a comprehensive prospective validation is warranted before these potential biomarkers are implemented in a clinical setting.

## Figures and Tables

**Figure 1 cancers-15-01448-f001:**
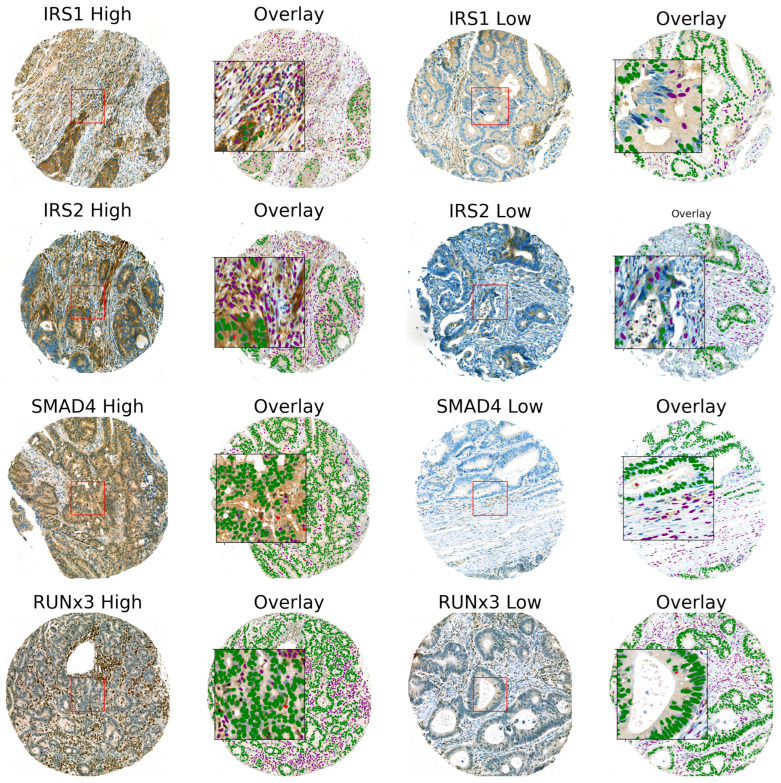
High and low expression of IRS1, IRS2, SMAD4, and RUNX3 in tumor and stromal cells. For SMAD4 and RUNX3, representative cores with high expression in both nucleus and cytoplasm were chosen.

**Figure 2 cancers-15-01448-f002:**
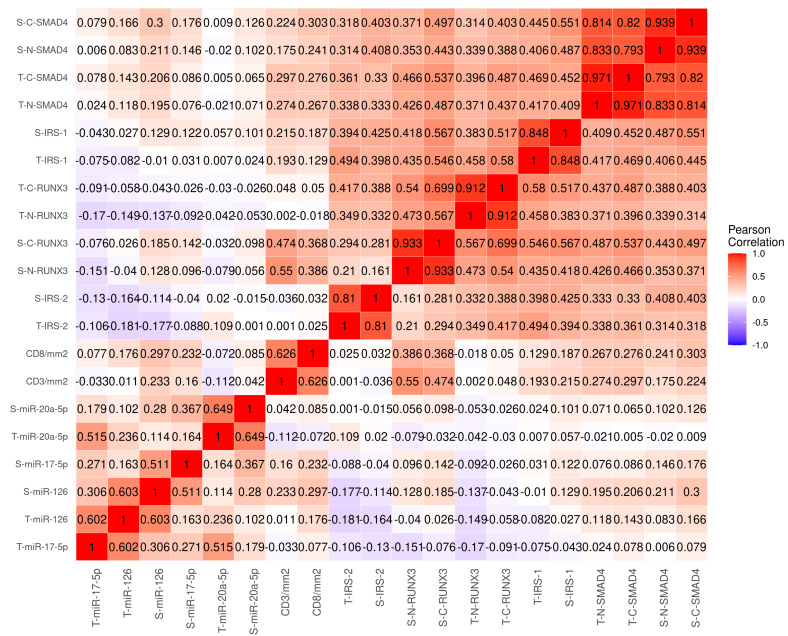
Correlations between IRS1, IRS2, SMAD4, and RUNX3 in tumor and stromal cell nucleus and/or cytoplasm, miR17, miR20a, and miR126 in tumor and stromal compartments and CD3+ and CD8+ cell density. Abbreviations: S, stroma; T, tumor; C, cytoplasm; N, nucleus.

**Figure 3 cancers-15-01448-f003:**
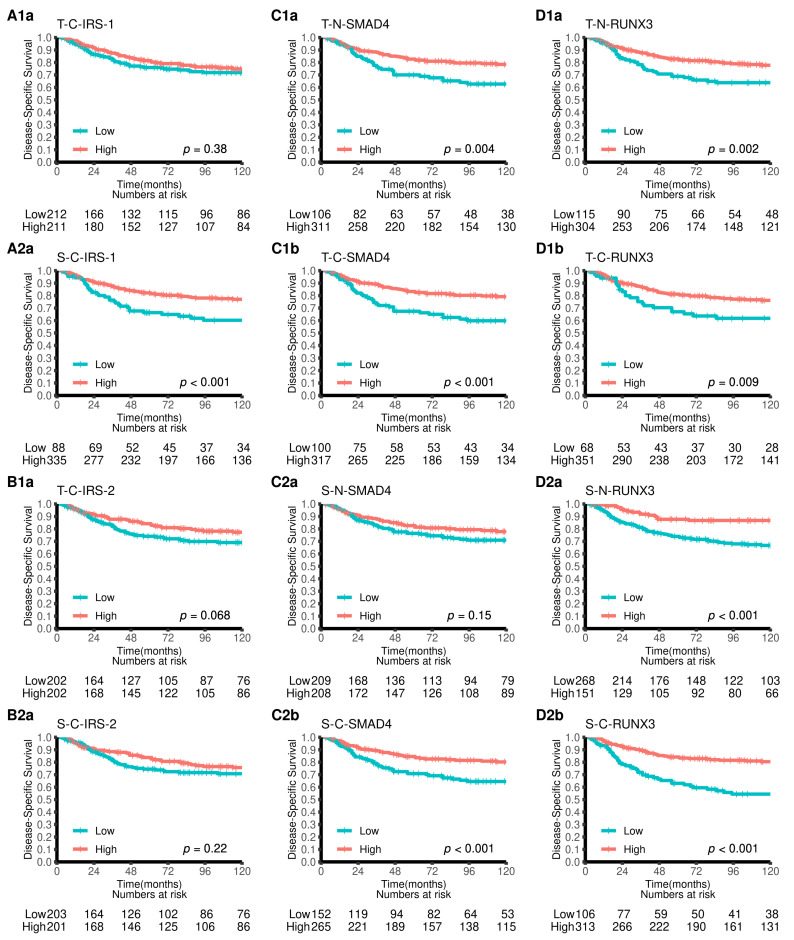
Disease-specific survival curves for the expression of IRS1 and IRS in tumor cell cytoplasm (**A1**–**A2**,**B1**–**B2**), SMAD4 (**C1**–**C2**), and RUNX3 (**D1**–**D2**) in tumor cell nucleus and cytoplasm using the optimal cutoffs for each marker. Abbreviations: S, stroma; T, tumor; C, cytoplasm; N, nucleus.

**Table 1 cancers-15-01448-t001:** Univariate analyses of tumor IRS1, IRS2, SMAD4, and RUNx3 in tumor and stromal cell nucleus and/or cytoplasm (log-rank test, *n* = 452).

	Tumor					Stromal				
	N(%)	5 Year	Median	HR(95% CI)	*p*	N(%)	5 Year	Median	HR(95% CI)	*p*
C-IRS1					0.380					<0.001
Low	212(47)	76	NA	1		88(19)	66	182	1	
High	211(47)	81	NA	0.84(0.57–1.24)		335(74)	82	NA	0.5(0.31–0.82)	
Missing	29(6)					29(6)				
C-IRS2					0.068					0.220
Low	202(45)	74	NA	1		203(45)	74	NA	1	
High	202(45)	84	NA	0.69(0.46–1.03)		201(44)	84	NA	0.78(0.52–1.16)	
Missing	48(11)					48(11)				
N-SMAD4					0.004					0.150
Low	106(23)	70	NA	1		209(46)	76	NA	1	
High	311(69)	83	NA	0.55(0.35–0.88)		208(46)	83	NA	0.74(0.5–1.11)	
Missing	35(8)					35(8)				
C-SMAD4					<0.001					<0.001
Low	100(22)	67	NA	1		152(34)	71	NA	1	
High	317(70)	83	NA	0.48(0.29–0.77)		265(59)	85	NA	0.49(0.32–0.74)	
Missing	35(8)					35(8)				
N-RUNX3					0.002					<0.001
Low	115(25)	69	NA	1		268(59)	74	NA	1	
High	304(67)	83	NA	0.53(0.34–0.83)		151(33)	88	NA	0.37(0.25–0.56)	
Missing	33(7)					33(7)				
C-RUNX3					0.009					<0.001
Low	68(15)	67	NA	1		106(23)	63	182	1	
High	351(78)	81	NA	0.55(0.32–0.95)		313(69)	84	NA	0.36(0.23–0.58)	
Missing	33(7)					33(7)				

Abbreviations: C, cytoplasm; N, nucleus; NA, not applicable.

**Table 2 cancers-15-01448-t002:** Multivariable models including statistically significant clinicopathological variables and investigated biomarkers from univariable analyses (Cox proportional hazards test, *n* = 452). Separate models for each marker in each compartment (tumor T1 and T2 and stromal S1–S3).

		Tumor					Stromal			
	T1		T2		S1		S2		S3	
	HR(95% CI)	*p*	HR(95% CI)	*p*	HR(95% CI)	*p*	HR(95% CI)	*p*	HR(95% CI)	*p*
Age	1.03(1.01–1.05)	0.005	1.02(1–1.05)	0.0141	1.02(1.01–1.04)	0.013	1.02(1–1.04)	0.020	1.02(1–1.05)	0.018
pTNM										
pTNM I	1		1		1		1		1	
pTNM II	1.7(0.66–4.42)	0.274	2.32(0.89–6.02)	0.083	1.89(0.73–4.9)	0.188	1.75(0.67–4.54)	0.252	2.17(0.83–5.63)	0.113
pTNM III	5.24(2.07–13.28)	<0.001	6.71(2.65–16.99)	<0.001	6.16(2.44–15.56)	<0.001	5.39(2.13–13.67)	<0.001	5.65(2.23–14.32)	<0.001
Margins										
0 mm	1		1		1		1		1	
<1 mm	0.58(0.26–1.27)	0.174	0.55(0.25–1.21)	0.135	0.62(0.28–1.38)	0.238	0.7(0.32–1.54)	0.375	0.49(0.22–1.07)	0.075
1–2 mm	0.17(0.05–0.54)	0.003	0.16(0.05–0.58)	0.005	0.27(0.1–0.78)	0.015	0.24(0.08–0.75)	0.014	0.16(0.05–0.57)	0.005
2–10 mm	0.33(0.16–0.69)	0.003	0.45(0.22–0.92)	0.028	0.45(0.22–0.92)	0.028	0.43(0.21–0.89)	0.024	0.39(0.19–0.79)	0.009
10–50 mm	0.44(0.22–0.86)	0.017	0.64(0.33–1.24)	0.185	0.55(0.28–1.07)	0.080	0.58(0.3–1.14)	0.112	0.53(0.28–1.02)	0.056
>50 mm	0.35(0.14–0.89)	0.028	0.41(0.17–0.99)	0.049	0.41(0.17–1.01)	0.054	0.42(0.17–1.09)	0.074	0.35(0.15–0.86)	0.022
C-IRS1										
Low					1					
High					0.64(0.47–0.87)	0.005				
N-SMAD4										
Low										
High		NS						NS		
C-SMAD4										
Low	1						1			
High	0.58(0.43–0.8)	<0.001					0.67(0.5–0.91)	0.009		
N-RUNX3										
Low			1						1	
High			0.62(0.45–0.84)	0.002					0.68(0.44–1.03)	0.068
C-RUNX3										
Low									1	
High				NS					0.62(0.44–0.87)	0.006

Abbreviations: S, stroma; T, tumor; C, cytoplasm; N, nucleus; NS, not significant.

## Data Availability

The data presented in this study are available on request from the corresponding author. The data are not publicly available due to privacy regulations.

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
