# Peer review of "High Expression of IRS-1, RUNX3 and SMAD4 Are Positive Prognostic Factors in Stage I–III Colon Cancer"

_cancers, 2023, doi:10.3390/cancers15051448_

Round 1

Reviewer 1 Report

Here, the authors examine the expression and prognostic impact of IRS-1, IRS-2, RUNx3 and SMAD4 in colon cancer, both in the epithelium and stroma.
The topic is original, and the study design is appropriate for the research question asked by the paper.
The methodology is rigorous and effective for characterizing the expression levels of these molecules within tissue samples.
The conclusions are consistent with the results of the study, and they are properly discussed and well-supported by easy to follow figures. The discussion is adequate, explaining the findings in the context of supporting literature and existing knowledge, highlighting the strengths and limitations of the study, and appropriate references have been cited.

This is a very interesting and novel study, only minor revisions are needed:

- lines 103-104: "The antibodies...were..." instead of "was"

- Results: please add P values in parentheses throughout the text.

- line 265: "...prognostic marker in CRC [?]" It seems that a reference is missing.

- As for RUNX3, it has been proposed that a further mechanism of action in colon cancer is the inhibition of Hedgehog signaling pathway doi: 10.1038/s41418-019-0379-5, please comment 

Author Response

We thank the reviewer for his/hers positive review of our paper “High Expression of IRS-1, RUNX3 and SMAD4 are Positive Prognostic Factors in Stage I-III colon cancer”. We hereby resubmit the revised manuscript after having carefully considered the remarks raised by the referee. Specific questions are answered below and changes in the manuscript have been made where necessary.

Lines 103-104: "The antibodies...were..." instead of "was": Changed in the text

Results: please add P values in parentheses throughout the text: To increase the readability of the text, we intentionally omitted P values from the correlation and univariate analyses. For correlation analyses, the sheer number of patients in the analyses gives P values < 0.001 for nearly all r values > 0.2 and all r values > 0.3. Hence the P value is implied. For univariate analyses, the referenced figures and tables are available next to the text and the most important results are referenced in the multivariable chapter with HRs and P values. 

Line 265: "...prognostic marker in CRC [?]" It seems that a reference is missing: The reference is now fixed in the text

As for RUNX3, it has been proposed that a further mechanism of action in colon cancer is the inhibition of Hedgehog signaling pathway doi: 10.1038/s41418-019-0379-5, please comment: We missed this novel study in our first round of literature review and thank the reviewer for pointing it out. A few lines summarizing Kim et als. findings and their implications for RUNX3 signalling in CRC have been added to the discussion

Yours sincerely
Thomas K Kilvaer

Reviewer 2 Report

In this manuscript, the authors have examined the expression level of IRS-1, IRS-2, RUNX3 and SMAD4 from colon cancer patients, especially in tumor epithelial and stromal compartments. With their analysis from digitized pathology images, they identified that stromal IRS-1, RUNX3 and SMAD4 are positive prognostic factors. Additionally, they pointed out that RUNX3 expression correlated with TILs density. Overall, the authors have concluded that IRS-1, RUNX3 and SMAD4 could serve as potential biomarkers. However, there are some remaining questions to be answered:

1, The authors have examined the protein expression levels in different cell types and localizations. Have the authors examined the total expression level? Are they correlated with any clinicopathological variables?

2, Since the cell type and localization matter, could the authors discuss their potential functions in tumor or stroma cells? In nucleus and cytoplasm? Do they have distinct functions in different cell types and locations? It is obvious that RUNX3 have opposite function within stroma nucleus and tumor cytoplasm in Grade.  

3, Could the authors include more rationales about why studying these proteins, including IRS-1 and 2, RUNX3 and SMAD4? The authors only mentioned that these are members thought to be regulated by these miRs from their previous works.

4, Some typo errors, including

line 265, [?] , missing reference.

Table S3, T-CRUNX3, should be T-C-RUNX3.

Author Response

We thank the reviewer for his/hers positive review of our paper “High Expression of IRS-1, RUNX3 and SMAD4 are Positive Prognostic Factors in Stage I-III colon cancer”. We hereby resubmit the revised manuscript after having carefully considered the remarks raised by the referee. Specific questions are answered below and changes in the manuscript have been made where necessary.

The authors have examined the protein expression levels in different cell types and localizations. Have the authors examined the total expression level?:The total expression levels were examined as per request of the reviewer. For IRS1, total expression level was highly correlated with both tumor (r > 0.8) and stromal (r > 0.9) expression. For IRS2, total expression level was highly correlated with both tumor (r > 0.8) and stromal (r > 0.8) expression. For RUNX3,, the total expression level was highly correlated with stromal (r > 0.8) and less so for tumor expression (r < 0.6). For SMAD4 total expression level was highly correlated with both tumor (r > 0.8) and stromal (r > 0.9) expression. These findings are consistent with our observations about the distribution of marker expression in section 3.2. first paragraph of the manuscript.

Are they correlated with any clinicopathological variables?:  Associations between clinicopathological variables and compartment specific expression is given in S3 Table. 

Since the cell type and localization matter, could the authors discuss their potential functions in tumor or stroma cells? In nucleus and cytoplasm? Do they have distinct functions in different cell types and locations? It is obvious that RUNX3 have opposite function within stroma nucleus and tumor cytoplasm in Grade.: We agree that further sub analyses within specific cell types would be highly interesting. However, as we state in section 2.4, limitations of the DL method obscures further fine grained cell level analyses. Fortunately, since the images are digitized, subsequent analyses can be conducted when even more sophisticated models are developed. Since we did not observe any large differences between nuclear and cytoplasmatic expression of the investigated markers, we chose not to pursue this issue in particular. Although, it would be an interesting topic for a systematic review.

Could the authors include more rationales about why studying these proteins, including IRS-1 and 2, RUNX3 and SMAD4? The authors only mentioned that these are members thought to be regulated by these miRs from their previous works: This work is part of a Selvens PhD thesis. As we already state in the manuscript, the biomarkers were chosen because  they were (supposedly) at least partly regulated by miRs 17-5p, 20a-5p and/or 126-5p. In addition,  we think that RUNX3 is an up-and-coming biomarker for cancer in general and for CRC in particular. To our surprise, no/only weak correlations between the biomarkers and miRs were observed. 

Table S3, T-CRUNX3, should be T-C-RUNX: Changed in the table

line 265, [?] , missing reference.: The reference is now fixed in the text

Yours sincerely
Thomas K Kilvaer